# Further Investigations of Nitroheterocyclic Compounds as Potential Antikinetoplastid Drug Candidates

**DOI:** 10.3390/biom13040637

**Published:** 2023-04-01

**Authors:** Carlos García-Estrada, Yolanda Pérez-Pertejo, Bárbara Domínguez-Asenjo, Vanderlan Nogueira Holanda, Sankaranarayanan Murugesan, María Martínez-Valladares, Rafael Balaña-Fouce, Rosa M. Reguera

**Affiliations:** 1Departamento de Ciencias Biomédicas, Facultad de Veterinaria, Universidad de León, Campus de Vegazana s/n, 24071 León, Spain; 2Medicinal Chemistry Research Laboratory, Department of Pharmacy, Birla Institute of Technology and Science Pilani, Pilani Campus, Pilani 333031, India; 3Instituto de Ganadería de Montaña (IGM), Consejo Superior de Investigaciones Científicas-Universidad de León, Carretera León-Vega de Infanzones, Vega de Infanzones, 24346 León, Spain; 4Departamento de Sanidad Animal, Facultad de Veterinaria, Universidad de León, Campus de Vegazana s/n, 24071 León, Spain

**Keywords:** nitroheterocycles, kinetoplastids, sleeping sickness, Chagas disease, leishmaniasis, drug discovery, drug repurposing

## Abstract

Due to the lack of specific vaccines, management of the trypanosomatid-caused neglected tropical diseases (sleeping sickness, Chagas disease and leishmaniasis) relies exclusively on pharmacological treatments. Current drugs against them are scarce, old and exhibit disadvantages, such as adverse effects, parenteral administration, chemical instability and high costs which are often unaffordable for endemic low-income countries. Discoveries of new pharmacological entities for the treatment of these diseases are scarce, since most of the big pharmaceutical companies find this market unattractive. In order to fill the pipeline of compounds and replace existing ones, highly translatable drug screening platforms have been developed in the last two decades. Thousands of molecules have been tested, including nitroheterocyclic compounds, such as benznidazole and nifurtimox, which had already provided potent and effective effects against Chagas disease. More recently, fexinidazole has been added as a new drug against African trypanosomiasis. Despite the success of nitroheterocycles, they had been discarded from drug discovery campaigns due to their mutagenic potential, but now they represent a promising source of inspiration for oral drugs that can replace those currently on the market. The examples provided by the trypanocidal activity of fexinidazole and the promising efficacy of the derivative DNDi-0690 against leishmaniasis seem to open a new window of opportunity for these compounds that were discovered in the 1960s. In this review, we show the current uses of nitroheterocycles and the novel derived molecules that are being synthesized against these neglected diseases.

## 1. Introduction

Estimates of the World Health Organization (WHO) about the prevalence and severity of the so-called neglected tropical diseases (NTDs) are conclusive and indicate that one-sixth of the world’s population is affected by at least one of these diseases, and it is not uncommon for them to occur together in parts of Africa and Asia. In addition to mortality, the high morbidity among individuals in endemic regions causes economic disruption due to disability-adjusted life years (DALYs), the cost of treatment and days of hospitalization, as these diseases can be fatal without medical intervention [1]. It should be noted that the incidence of these diseases most severely affects the most vulnerable members of the population in endemic countries, such as children and women. Last but not least, the suffering caused to families whose income is severely compromised by these diseases, either by the cost of treatment or by the loss of income during illness, can be as important as the epidemiological data.

NTDs caused by parasitic protozoa are among the most prevalent ones, and only appropriate public health and hygiene measures can partially mitigate them. However, in the absence of an effective preventive vaccine, a pharmacological approach is the only useful tool that can control these diseases when they occur [2]. The family Trypanosomatidae includes some of the single-celled parasites responsible for some of the most important NTDs in certain regions of low-income countries in Africa, Asia and South America, where little or no health care is available. According to WHO [1], three of the world’s major NTDs are human African trypanosomiasis (HAT) or sleeping sickness, caused by *Trypanosoma brucei* (*rhodesiense* and *gambiense* subspecies) endemic in Central and East African countries; American trypanosomiasis or Chagas disease, caused by *Trypanosoma cruzi* in South American countries, and recently emerging in Europe due to migratory flows; and, finally, diseases known generically as leishmaniasis, caused by several species of the genus *Leishmania* in several continents and responsible for a different pathophysiology of varying severity [3]. 

### 1.1. Human African Trypanosomiasis (HAT)

HAT, commonly called sleeping sickness, is a zoonotic disease caused by different subspecies of the bloodstream parasite *T. brucei* and transmitted by the Glossina tsetse fly [4]. The incidence of HAT is clearly decreasing thanks to health efforts in the target countries. Case statistics reported by WHO over the last 10 years show a clear decline in the total incidence of HAT (both *gambiense* and *rhodesiense* forms), which decreased by 97% (2009–2018 period), with the incidence in 2018 being 977 cases [5] and only 663 (the *gambiense* form accounts for more than 95% of these cases) in 2020 [6]. The estimated DALYs in 2010 were 560,000, 72% less than the 1990 statistics [7], with the estimated value of DALYs in 2017 being 79,000 [8]. However, 55 million people remain at risk of the disease in 36 countries in sub-Saharan Africa (the Democratic Republic of the Congo contains about 70% of this population) (https://www.who.int/news-room/fact-sheets/detail/trypanosomiasis-human-african-(sleeping-sickness); accessed on 3 March 2023).

The disease progresses in two stages, a so-called hemolymphatic phase characterized by episodes of fever, headache, lymphadenopathy, joint pain and itching, and a more severe neurological phase in which the parasites cross the blood–brain barrier and infect the central nervous system. Signs and symptoms of the disease include: behavioral changes, confusion, sensory disturbances, lack of coordination and sleep phase disturbances, which are serious and can be life-threatening. The severity of HAT depends on the parasite sub-species involved in the infection. *T. brucei gambiense* causes chronic infection in Central and West Africa and is responsible for more than 95% of reported cases. The disease may remain unnoticed for months or years, and by the time neurological symptoms appear, the disease stage is very advanced. However, *T. brucei rhodesiense* causes acute infection in southern East Africa and is responsible for less than 3% of the total cases [9]. 

### 1.2. American Trypanosomiasis (Chagas Disease)

*T. cruzi* is the trypanosomatid responsible for the vector-borne zoonotic disease called American trypanosomiasis or Chagas disease. This pathogen is transmitted through the contaminated feces of the triatomine *Triatoma infestans* when it penetrates through mucous membranes or open wounds in human skin [10,11]. In some countries, oral transmission of Chagas disease has been proposed, the primary vehicle being food and beverages contaminated with whole infected triatomines or their feces, including metacyclic trypomastigotes of *T. cruzi* [12]. Chagas disease is restricted to the South American subcontinent, but it is considered an emerging disease in non-endemic Southern European countries due to migratory flows from endemic Latin American countries in recent decades [13]. According to WHO, the number of cases of Chagas disease worldwide has experienced a significant reduction, from more than 30 million in 1990 to 6–8 million in 2010 [14]. However, despite these favorable figures, nearly 10,000 people die each year from complications related to the disease, and 232,100 estimated DALYs are lost due to this disease, which represents only a 2.8% reduction in the period 2007–2017 [8].

Despite its severity, 70–80% of people infected with *T. cruzi* remain asymptomatic throughout their lives. In those developing the disease, it progresses in two clinical phases: an acute phase and a chronic phase [15]. The acute phase occurs with a high parasitemia and mild signs of headache, fever, joint pain, lymphadenopathy, etc. However, the chronic phase of the disease, developed by 20–30% of infected persons, is invasive and responsible for cardiac [16] and digestive [17] disorders that can be fatal even when treated.

### 1.3. Leishmaniasis

Single-celled parasites of the genus *Leishmania* cause a group of vector-borne zoonotic and anthroponotic diseases commonly known as leishmaniasis, all of which are transmitted by *Phlebotomus* and *Lutzomya* sandflies worldwide [3]. The mildest presentation, cutaneous leishmaniasis (CL), causes ulcerations and papules around the site of the insect bite, and although it is not life-threatening, it is disfiguring, producing permanent scarring and being the source of stigma and social exclusion [18]. CL is mainly caused by *L. major* and *L. tropica* in the Old World and by *L. braziliensis* complex in the New World [19]. However, the mucocutaneous form of the disease (MCL), caused by the *L. amazonensis* complex in the New World, is a more severe and stigmatizing presentation of the disease, as the infection progresses from a simple sore at the bite site to the complete destruction of the mucous membranes of the mouth and nose [20]. Finally, visceral leishmaniasis (VL) is produced by *L. donovani* and *L. infantum* in the Old World and by *L. chagasi* (*infantum*) in the New World. In this case, the infection is systemic, invading internal organs, such as the thymus, liver, spleen and bone marrow, and producing renal dysfunction [21]. VL is fatal when patients are not pharmacologically treated and, in case of healing, it may evolve into a post-kala azar dermal leishmaniasis (PKDL), a rare form of skin disease that occurs with some frequency following antimony therapy [22]. The current incidence of VL is 50,000 to 90,000 new cases every year, whereas for the different forms of cutaneous leishmaniasis, this value ranges from 600,000 to 1 million (https://www.who.int/news-room/fact-sheets/detail/leishmaniasis; accessed on 3 March 2023), with more than 5700 deaths in 2019 according to the Drugs for Neglected Diseases initiative (DNDi) (https://dndi.org/diseases/visceral-leishmaniasis/facts/; accessed on 3 March 2023), although these values likely represent the tip of the iceberg, as the paucity of data from unstable contexts may contribute to inaccurate disease estimates. Eradication campaigns in the Indian subcontinent have reduced the incidence of the disease in India, Nepal and Bangladesh by more than 50% [23]. This may have contributed to the decline in the DALYs rate for visceral leishmaniasis in the period 1990–2017, which was estimated at 775,000 (511,000 for visceral and 264,000 for cutaneous leishmaniasis) [8].

## 2. Current Treatments against Trypanosomatids

Despite the host immune response playing an important role in healing and relapse prevention, there is no effective vaccination against these parasitic diseases in humans. Therefore, drug therapy is the only curative and preventive measure against them. However, many of the drugs used against NTDs are outdated, associated with adverse side effects and/or therapeutic failures due to the presence of resistant strains developed over years of exposure. Despite these drawbacks, we should not forget that these drugs have saved hundreds of thousands of lives throughout history and have been included as part of the WHO Essential Medicines List (https://www.who.int/groups/expert-committee-on-selection-and-use-of-essential-medicines; accessed on 3 March 2023).

### 2.1. Current Drug Treatments against American Trypanosomiasis

Two nitroheterocyclic compounds benznidazole and nifurtimox (Figure 1), approved for the pharmacological treatment of acute and chronic presentations of Chagas disease [24,25,26], have been used for more than five decades. Although benznidazole is more widely used than nifurtimox, both drugs appear to have similar efficacy and safety profiles. 

The nitroimidazolic prodrug benznidazole (N-benzyl-2-(2-nitro-1H-imidazol-1-yl) acetamide) (Figure 1a) was introduced as a first-line treatment in the mid-1970s against acute and chronic stages of Chagas disease [27]. In vitro studies have suggested that susceptibility to benznidazole depends on different *T. cruzi* strains, though IC_50_ values ≤19.5 μg/mL (75 μM) with 10-fold variations have been determined within the same assay, the trypanocidal effect being rapid and dependent on time and concentration [28,29,30]. Novel metal (Cu and Ag)-benznidazole complexes have been recently synthesized and tested against *T. cruzi* epimastigotes and amastigotes, with Ag-benznidazole complexes showing better activity than benznidazole alone and showing higher selectivity towards *T. cruzi* parasites [31].

Benznidazole has good oral bioavailability, following a monocompartmental pharmacokinetic pattern with a terminal half-life of 12.1 h [32]. The current recommended administration of benznidazole is 5–7 mg/kg body weight orally divided into 2–3 daily doses for 60 days in adults [33,34]. Since half-life is shorter in children, dose adjustment is required to gain effectiveness [35]. On August 29, 2017, the FDA approved benznidazole for use in children from 2 to 12 years old (https://www.fda.gov/newsevents/newsroom/pressannouncements/ucm573942.htm; accessed on 3 March 2023), and the treatment regime includes a child-adapted dose of 12.5 mg per tablet twice daily for 60 days (https://dndi.org/research-development/portfolio/paediatric-benznidazole/; accessed on 3 March 2023). Several studies have been conducted to assess treatment responses in the chronic indeterminate phase in children [36], concluding that the earlier children are treated, the better the response achieved [35,37,38,39]. Other clinical trials have been performed to evaluate the effect of benznidazole in the treatment of chronic indeterminate Chagas disease in adults. In a prospective, randomized clinical trial (NCT01162967) of adults with chronic *T. cruzi* infection conducted to test the effect of posaconazole (400 and 100 mg) and benznidazole (150 mg) given twice daily for 60 days, significantly more patients in the posaconazole groups than in the benznidazole group had treatment failure during follow-up [40]. The possibility of combining benznidazole (200 mg) with posaconazole (400 mg) was studied in a prospective, multicenter, randomized clinical trial (NCT01377480) in asymptomatic patients with Chagas disease in Latin America and Spain, concluding that benznidazole monotherapy was superior to posaconazole and to combination treatment with both drugs [41]. Another phase 2 clinical trial (NCT01489228) in Bolivia in a group of adults diagnosed with indeterminate Chagas disease tested the safety and efficacy of three oral regimens of ravuconazole and benznidazole. Benznidazole was administered for 60 days at 5 mg/kg daily, with a rapid and sustained effect on parasite clearance, and 82% of patients showed a sustained response at 12-month follow-up [42].

Although benznidazole treatment is effective and well tolerated and helps prevent vertical transmission of the parasite to the developing fetus [43], it has some limitations, such as long treatment duration, safety and tolerability issues, and insufficient efficacy in chronic cases of the disease [34]. The most frequent side effects are allergy, dermopathy, nausea and vomiting. The least frequent are polyneuropathy and bone marrow depression [44].

To explore exposure reduction in order to improve tolerability while maintaining efficacy, DNDi decided to test the efficacy of new benznidazole regimens. The BENDITA (Benznidazole New Doses Improved Treatment & Therapeutic Associations) study sought new dosing regimens that were at least as effective as standard treatment, with fewer side effects. In a phase 2 clinical trial in Bolivia (NCT03378661) in adults with indeterminate chronic Chagas disease, benznidazole (150–300 mg daily) as monotherapy or in combination with fosravuconazole was well tolerated and induced an effective antiparasitic response, regardless of treatment duration (2–8 weeks) [45]. Other trials are underway to evaluate the efficacy of different benznidazole regimens in patients with chronic Chagas disease, including MULTIBENZ (NCT03191162) in adult patients [46] and the safety and efficacy study in women of reproductive age BETTY (NCT03672487) [47], whose results will provide valuable information on the current treatment of the disease. DNDi is currently conducting a phase 3 clinical trial in Argentina to validate a two-week benznidazole regimen for adults in the chronic phase of the disease. (https://dndi.org/research-development/portfolio/new-benz-regimens/; accessed on 3 March 2023). These studies may change treatment regimens in the next few years.

Nifurtimox ((RS)-3-methyl-N-[(1E)-(5-nitro-2-furyl)methylene]thiomorpholin-4-amine 1,1-dioxide) (Lampit, Bayer HealthCare AG) is a sulfone-containing 5-nitrofuran (Figure 1b) that has been used for more than five decades, mainly as a second-line option in the treatment of Chagas disease [48]. Acute infections are treated with nifurtimox by oral administration of 8–10 mg/kg body weight in adults or 15–20 mg/kg body weight in children, in 2–3 daily doses for 60 days [49]. Adverse side effects of nifurtimox are common and include anorexia, vomiting, gastric pain, insomnia, headache, myalgia and seizures [50]. Nifurtimox is well-absorbed orally, but it suffers extensive first-pass hepatic metabolism. Administration of a single dose of 15 mg/kg resulted in a peak plasma concentration at 2 h, with a Cmax of 751 ng/mL and a terminal half-life of 2.95 h [51]. The low concentrations of Nifurtimox excreted in breast milk support the hypothesis that maternal treatment during lactation may be advantageous for nursing infants [52]. Several clinical trials with nifurtimox have been conducted over the past 5 years to find safer regimens in more vulnerable populations. CHICAMOCHA 3 (EQUITY) (NCT02369978) is a prospective clinical trial designed to evaluate the efficacy and safety of nifurtimox and benznidazole against American trypanosomiasis in Colombia and Argentina. The design includes the administration of 480 mg/day nifurtimox or 300 mg/day benznidazole for 60 days (full dose) or 240 mg/day nifurtimox or 150 mg/day benznidazole for 120 days (half dose) to patients who tested positive for *T. cruzi* without clinical signs of dilated cardiomyopathy. The EQUITY trial will report on the trypanocidal effect and the equivalence of these two compounds, and the results provided by this trial may challenge current recommendations on the choice of these agents [53]. Recently, a prospective phase 3 clinical trial (NCT02625974, CHICO) was performed in South America to evaluate the efficacy and safety of a new pediatric formulation of nifurtimox for children up to 17 years old with Chagas disease. Nifurtimox was administered as divisible, dispersible 30 mg and 120 mg tablets in children, who were randomized 2:1 to receive nifurtimox 10–20 mg/kg/day (aged <12 years) or 8–10 mg/kg/day (aged ≥12 years) for 60 days or for 30 days plus placebo for 30 days. With an excellent therapeutic response, serological values were lower for the 30-day treatment than for the 60-day treatment, and the treatment led to a substantial decrease in the frequency of *T. cruzi*-positive PCR results. The 60-day treatment gave rise to a serological response 1 year after treatment that was superior to the placebo group and was well tolerated, with a favorable safety profile in infected children [54]. A second part of this study, CHICO SECURE, is a 3-year long-term follow-up to assess the incidence of seronegative conversion in children with a diagnosis of Chagas disease who were randomized and received at least one dose of their assigned 60- or 30-day nifurtimox treatment regimen. This portion of the study was designed as a post-marketing commitment at the request of the FDA.

### 2.2. Current Drug Treatments against HAT

The clinical drugs currently used against HAT (Figure 2) are scarce, and not all of them are effective against the late neurological phase of the disease. 

When HAT is detected in the early hemolymphatic phase, pentamidine and suramin are quite effective, but not when the parasite has penetrated the blood–brain barrier. Only the arsenical derivative melarsoprol and the fluorinated analogue of L-ornithine eflornithine are then effective. Two nitroheterocyclic drugs, fexinidazole and nifurtimox (in combination with eflornithine), have recently been introduced in the treatment of HAT.

Pentamidine isethionate is an aromatic diamidine (Figure 2a) prescribed only for the early non-neurological stage of *gambiense*-HAT, not for intermediate or late stages of the disease [55]. Due to its cationic nature, pentamidine has low oral bioavailability and should be administered by intramuscular injection at 4 mg/kg body weight at 24 h intervals for one week [56]. Under this regimen, pentamidine is generally well tolerated but may cause pain at the injection site, vomiting, hypotension, tachycardia and skin irritation [57].

The old drug suramin is a polysulfonated napthylamine derivative (Figure 2b) that was introduced for the treatment of HAT in 1922 and is preferentially effective against non-neurological early stages of rhodesiense-HAT [58]. Suramin is poorly absorbed by the oral route and has to be intravenously administered at a dose of 4–5 mg/kg body weight on day 1, followed by five injections of 20 mg/kg body weight on days 3, 10, 17, 24 and 31 (maximum dose per injection: 1 g) [9]. Suramin has several major adverse side effects, including hypersensitivity, nephropathy, peripheral neuropathy and bone marrow toxicity [59].

For the treatment of the neurological stage of *gambiense*-HAT, melarsoprol, an arsenite-based drug, was extensively used for years until the introduction of eflornithine, a more effective and user-friendly drug, but remains the only option for treatment of rhodesiense-HAT [60]. Melarsoprol (Figure 2c) is actually a prodrug that has to be metabolized to an active form of As^3+^ (melarsen-oxide), which penetrates the CNS. Melarsoprol is the drug of choice against *gambiense* HAT when eflornithine therapy is not affordable. The drug is highly effective when administered in a 10-day intravenous infusion schedule consisting of 2.2 mg/kg body weight/day (over 93.9% cure rate) [61]. Melarsoprol is a very unsafe drug that requires hospitalization due to the high rate of adverse side effects, such as encephalopathy and metastatic cascade [62].

The introduction of eflornithine was a major breakthrough in HAT treatment [63]. Eflornithine (Figure 2d) and other L-ornithine analogues were originally designed to target ornithine decarboxylase (ODC), the key enzyme of polyamine biosynthesis in proliferative processes [64]. As a consequence, it was soon used as first-line treatment for second stage *gambiense*-HAT, but not against rhodesiense-HAT due to the lack of innate susceptibility of this parasitic form [65]. Eflornithine is a water-soluble drug that requires administration by slow intravenous infusion. Due to the low oral bioavailability and short half-life of eflornithine, it is necessary to administer it in large intravenous infusions (100 mg/kg body weight every 6 h for 14 days) to obtain a 99% cure rate. Eflornithine is a water-soluble drug that requires administration by slow intravenous infusion. Due to the low oral bioavailability and short half-life of eflornithine, it is necessary to administer it in large intravenous infusions (100 mg/kg body weight every 6 h for 14 days) to obtain a 99% cure rate. These pharmacokinetic problems are aggravated by the high cost of the drug, which not all African countries can afford [9]. Although eflornithine is considered a very safe drug, some side effects, such as diarrhea, dizziness, headaches, alopecia and convulsions, have been reported. [66].

Recently, an orally administered nitroheterocyclic derivative, fexinidazole (1-methyl-2-((p-(methylthio)-phenoxy)methyl)-5-nitroimidazole) (Figure 2e), has been introduced in some African countries, having been approved by the FDA and received a positive opinion from the EMA as the first all-oral therapy for the hemolymphatic and meningoencephalic forms of HAT [67]. In field conditions, fexinidazole has been approved in the Democratic Republic of the Congo as a 10-day treatment for *gambiense*-HAT (https://dndi.org/press-releases/2019/fexinidazole-sleeping-sickness-approved-democratic-republic-congo/; accessed on 3 March 2023). 

Fexinidazole was developed by Hoechst AG (now part of Sanofi) in the 1970s and early 1980s as a broad-spectrum antimicrobial agent [68,69]. At that time, in vivo activity of fexinidazole against chronic *T. brucei* infections was observed [70] but was not further developed. Later, DNDi rediscovered it as a promising drug candidate for the treatment of HAT, after an extensive compound search among more than 700 new and existing nitroheterocycles [71,72,73]. After oral administration, fexinidazole is metabolized by CYP450 and FMN enzymes to two metabolites: fexinidazole sulfoxide and fexinidazole sulfone, the latter being eliminated more slowly [74]. Watson and co-workers, in a retrospective study using all pharmacokinetic data available to date, concluded that fexinidazole follows a monocompartmental disposition model with two transit compartments for the formation of the sulfone, which has an apparent volume of distribution of 80 L and an oral clearance of 3 L/h [75]. Both sulfoxy and sulfone actively killed different *T. brucei* strains (*T. b. gambiense*, *T. b. rodhesienese* and *T. b. brucei*) in vitro in the 0.7 to 3.3 μM range. Preclinical studies concluded that fexinidazole was effective in acute in vivo models of *T. b. gambiense* and *T. b. rhodesiense* at a dose of 100 mg/kg of body weight/day for 4 days and in chronic infections of *T. b. brucei* at 200 mg/kg/day for 5 days [71,76]. Fexinidazole was confirmed to have a dose-dependent effect (range of 7.4 to 200 mg/kg daily for 5 days), as relapses occurred early at the lower doses of 22 mg/kg and 66.7 mg/kg in a mouse model of chronic infection with a *T. b. brucei* bioluminescent strain [77]. Clinical trials with fexinidazole concluded that oral monotherapy was effective against *gambiense* HAT. In a randomized, multicenter, phase 2/3 trial (NCT01685827), patients aged ≥ 15 years with stage-2-*gambiense* HAT were treated once daily for 10 days with oral fexinidazole in a regimen of 1800 mg (3 × 600 mg tablets) on days 1–4 followed by 1200 mg (2 × 600 mg tablets) on days 5–10. After 18 months, the success rate, which was defined as survival to treatment linked to parasitological clearance in body fluids, with no need for rescue medication and with ≤ 20 white blood cells/μL of cerebrospinal fluid, was 91.2% in patients treated with fexinidazole [78]. In another prospective, multicenter study (NCT02169557), patients aged ≥ 15 years with stage 1 or early stage 2 *gambiense*-HAT received daily 1800 mg fexinidazole on days 1–4 followed by 1200 mg fexinidazole on days 5–10, with treatment being effective at 12 months, with a success rate of 99% [79]. Several field studies have been carried out to assess the potential pediatric use of fexinidazole. A prospective, multicenter pivotal study [NCT02184689] reported a treatment success of 97.6% 12 months after completing fexinidazole treatment (600 mg tablets) in children aged 6–14 years weighing over 20 kg with *gambiense*-HAT at any stage [80]. Another clinical phase 3 trial in patients aged ≥ 6 years with *gambiense*-HAT of any stage treated (NCT03025789), including breastfeeding or pregnant women in the second or third trimester, was initiated on November 17 2016. Later, fexinidazole received a positive opinion from the EMA for treatment of both first stage (hemolymphatic) and second-stage (meningoencephalitic) HAT caused by *T. b. gambiense* in adults and children 6 years and older and weighing 20 kg or more. This is the first oral regimen for *gambiense*-HAT that is effective in treating both disease stages [81].

More recently, a phase 2/3 study has included fexinidazole to treat rhodesiense-HAT in Malawi. Unfortunately, the study was put on hold due to the COVID-19 pandemic. According to the DNDi website, the study will continue until the end of 2022 to allow for the follow-up of all patients for 12 months (https://dndi.org/research-development/portfolio/fexinidazole-tb-rhodesiense/; accessed on 3 March 2023).

In conclusion, regarding these studies, the EMA indicates that fexinidazole, in the form of 600 mg tablets, should be taken once daily with food for 10 days. Patients weighing 35 kg or more should take 3 tablets a day for the first 4 days and 2 tablets a day for the next 6 days. For patients weighing 20–35 kg, the daily dose should be 2 tablets for the first 4 days and 1 tablet for the following 6 days. Treatment should be supervised by trained healthcare staff to ensure that patients take the tablets according to their needs [73]. The FDA approved fexinidazole as the first all-oral treatment for both phases of sleeping sickness in 2021 (https://dndi.org/press-releases/2021/us-fda-approves-fexinidazole-as-first-all-oral-treatment-sleeping-sickness/; accessed on 3 March 2023).

Before the introduction of fexinidazole, the most advanced therapy against *gambiense*-HAT was the combination of eflornithine (Figure 2d) with the nitroheterocyclic antichagasic compound nifurtimox (Figure 1b) [82]. Nifutimox, Eflornithine Combination Therapy (NECT) became the first-line treatment for second stage HAT after its inclusion in the WHO Essential Medicines List and the WHO Essential Medicines List for children in 2009 and 2013, respectively [83,84]. NECT consists of the oral administration of nifurtimox along with intravenous administration of eflornithine. The efficacy and safety of NECT for second-stage disease compared with the standard eflornithine regimen were assessed in patients aged 15 years or older in a multicenter phase 3 trial (NCT00146627) in the Republic of the Congo and the Democratic Republic of the Congo. In this study, it was shown that the efficacy of NECT was non-inferior to eflornithine monotherapy. In addition, NECT was easier to administer, presented safety advantages and was potentially protective against the emergence of resistant parasites [85]. Under these conditions, NECT reduces the duration of treatment by almost 50% with only 14 total slow infusions of 400 mg/kg eflornithine administered every 12 h for 1 week and a concurrent 10-day oral treatment with 15 mg/kg of nifurtimox in three doses administered orally to achieve a cure rate of over 97% [85,86]. NECT continues to be the first-choice treatment for patients aged below 6 years with severe second-stage HAT (≥100 white blood cells/μL in the cerebrospinal fluid) or patients with second-stage HAT (>5 white blood cells/μL in the cerebrospinal fluid) and weighing less than 20 kg [87]. In addition, the NECT kit for full chronic treatment of CNS *gambiense* infection was shown to be more cost-effective than using eflornithine alone [88,89]. An additional phase 3b clinical trial of NECT efficacy with a wide population range, including children and pregnant and breastfeeding women suffering chronic stage *gambiense* sleeping sickness, was carried out in the Democratic Republic of the Congo (NCT00906880), showing that NECT was tolerable, effective and appropriate for use in a broad population, including vulnerable subpopulations [90,91].

### 2.3. Current Drug Treatments against Leishmaniasis

Four active substances (Figure 3) are currently used in monotherapy or in combination for the management of the different forms of leishmaniasis, with each drug, formulation or administration schedule being adapted to the presentation of the disease and the characteristics of the geographical area in which the outbreak has occurred [3].

The pentavalent antimony derivatives (Sb^V^), meglumine antimoniate (Glucantime) (Figure 3a) and sodium stibogluconate (Pentostam) (Figure 3b) remain the first-line drugs used against leishmaniasis both as monotherapy and in combination with paromomycin in certain regions of Africa [92]. Sb^V^-based drugs are in fact prodrugs (Sb^V^ must be reduced to Sb^III^ to be effective) that have been marketed since the mid-1930s, and due to their continued use, resistant strains of *Leishmania* sp. have emerged in certain regions of the world [93]. Sb^V^ is administered by intramuscular or subcutaneous injections at 20–30 mg/kg body weight/day for 25–30 days for VL [94]. Multiple intralesional injections are recommended for CL in Latin American countries [19]. In addition, the poor adherence to the Sb^V^ regimen, due to the high number of injections, the pain of the injections and the multiple side effects, can lead to relapses and the development of the disfiguring PKDL form [95]. Sb^V^-based drugs have some common toxic adverse effects: arthralgia, myalgia, hepatotoxicity, pancreatitis and nephrotoxicity [96]. Cardiotoxicity is a serious side effect of these drugs, and therefore their prescription is recommended to young people over 15 years of age and adults up to 45 years of age [97].

The polyene macrolide antifungal amphotericin B (AmpB) (Figure 3c) is marketed as a deoxycholate salt (Fungizone) or as a liposome formulation (AmBisome). The poor oral bioavailability of AmpB deoxycholate and AmBisome implies that they have to be administered by slow intravenous infusion. Intravenous infusion of AmpB at 0.75–1 mg/kg body weight/day for 15–20 days is almost 100% effective against refractory VL in the Indian subcontinent [72]. However, this form of administration involves the patient’s hospitalization during the long course of treatment. Alternatively, a single injection of AmBisome at 10 mg/kg body weight has been introduced with a remarkable cure rate of 96% and is becoming the gold standard in India, where Sb^V^ resistance is highly prevalent [98,99]. AmpB formulations are extremely effective drugs, but there are serious management problems, such as poor oral bioavailability, poor chemical stability in extreme climatic conditions and unacceptable cost in target countries [100]. AmpB is not a completely safe drug, and adverse effects, such as nephrotoxicity, hypokalemia and myocarditis, are common. These adverse effects are greatly reduced with the use of AmBisome [101].

The alkylphospholipid miltefosine (Figure 3d) is a repurposed drug designed and developed as an oral antitumor drug but with remarkable antileishmanial characteristics. Miltefosine is a second-line drug that, when administered orally at 50–100 mg/kg body weight/day for a total of 28 days, yields a cure rate of almost 95% [102]. Part of its efficacy is due to its long half-life (more than 1 week), which results in a 5- to 10-fold increase in plasma levels under stable conditions [103]. However, the antileishmanial efficacy of miltefosine is much lower in children on a weight-dependent schedule, and allometric administration should be introduced to increase its efficacy to adult levels [104]. Despite its safety, miltefosine causes adverse intestinal cramps, vomiting, diarrhea and anorexia. In addition, suspected teratogenic problems with miltefosine make its use during pregnancy contraindicated [105].

Finally, the aminoglycoside antibiotic paromomycin (Figure 3d) has been shown to be highly effective against VL in combination with other antileishmanial drugs. Paromomycin has poor oral bioavailability and only intramuscular injections (11 mg/kg body weight/day for 21 days) have been shown to reduce parasite burden in cases of VL in India, but not in East Africa [106]. The combination of pentostam + paromomycin in East Africa reduces the treatment time for Sb^V^ and improves patient compliance. In addition to painful intramuscular injections, other reported adverse effects include nephrotoxicity and ototoxicity [107].

### 2.4. Drug Resistance to Current Pharmacology

The lack of a varied offer of drugs against trypanosome diseases and their overuse is at the origin of their loss of efficacy and the appearance of resistances. Big pharmaceutical companies have made little response to this situation due to the neglected nature of these diseases. However, it should not be forgotten that all parasites—which are in a constant arms race with the host—develop strategies to escape both the host’s immune system and the drugs administered [108]. These resistances are sometimes generic, such as systems related to drug absorption and elimination, and sometimes specific to a particular drug. To these resistance processes, we must add the pathogen′s ability to persist in dormant states—persisters—of low metabolism and lower sensitivity to drug treatment [109]. Finally, we must mention the lack of clear therapeutic targets in trypanosomatids, which hinders and delays the discovery and development of new therapeutic entities against these diseases [110]. Although it is not the purpose of this review, we will briefly introduce some examples of drug resistance to trypanosomatids. Among the mechanisms of generic resistance to antileishmanial drugs, the case of Sb^V^ derivatives is well known. 

For years, Glucantime has been known to be therapeutically ineffective in the state of Bihar in Northeast India [111]. Resistance is associated with drug transport due to mutations in aquaglyceroporin 1 that have been linked to high levels of arsenic in drinking water [112]. Similarly, mutation or loss of aquaglyceroporin 2 is also implicated in the resistance of certain strains of *T. b. gambiense* to melarsoprol [113]. Resistance to miltefosine—easily reproducible in the laboratory by successive passes at increasing drug concentrations—has been attributed to mechanisms of reduced drug absorption under field conditions [114].

In addition to the alteration of transport mechanisms, there have been cases of resistance linked to drugs with known mechanism of action, as is the case of eflornithine, the ODC inhibitor. In addition to mutations in the amino acid transporter AAT6 [115], *T. b. brucei* strains with silenced ODC and compensated with an upregulated putrescine transport mechanism were used to explain resistance to this drug, a component of NECT therapy [116,117]. In the case of nitro compounds, such as nifurtimox (the second component of NECT therapy), resistant strains related to the prodrug nature of these compounds have also been described. Nifurtimox is a prodrug that must undergo two reductions in its nitro group catalyzed by type I NTRs, directly forming intermediate species toxic to the parasite [118]. Low levels of NTR I of trypanosomes directly affect their sensitivity to nifurtimox and other nitro derivatives, since reduced levels of the enzyme lead to cross-resistance to other nitro compounds, such as benznidazole and fexinidazole [119]. In the case of AmpB, cases of resistance are not very frequent and have been associated with mutations in sterol metabolism genes (sterol methyl transferase or the sterol C5 desaturase) [120].

Persistent phenotypes—a transient subpopulation that is less susceptible to drug treatment and may remain in a dormant state after drug treatment—have also been described in trypanosomatids. In Chagas disease, both the long-term persistence of parasites in patients and the frequent failure of standard therapies (benznidazole and nifurtimox) have been reported [121]. There is also evidence of persister forms in leishmaniasis. PKDL is an example of relapse occurring after treatment and apparent cure of VL with antimonials in India [122,123].

The evolution of drug resistance is a serious problem, since it makes current treatments less potent as they are administered over time. Therefore, new strategies are necessary to slow the rate of resistance to current treatments against diseases caused by trypanosomatids. One of these strategies is drug combination, which comprises the use of at least two drugs with different mechanisms of action. Those drugs must be carefully selected to minimize the acquisition of resistance in order to improve clinical outcomes [124].

## 3. New Drugs Entering Clinical Trials against Trypanosomatid-Borne Diseases

The current clinical pharmacology of NTDs in endemic countries is limited to the drugs listed above. In many cases, their efficacy has declined over time, while other compounds are either economically unaffordable for low-income countries or chemically unstable. In addition, most of them require repeated parenteral administration over long periods to be effective, which compromises their adherence. DNDi predicted, after the 2012 London Declaration on NTDs (https://www.gov.uk/government/publications/london-declaration-on-negelected-tropical-diseases-ntds; accessed on 3 March 2023), effective treatments against these diseases in 2020 [125], but no major progress has been made in this field, apart from the implementation of new combination therapies for VL and some trials in phase 3 for HAT and Chagas disease. However, the popularization of automated HTS screening methodologies aided by bioimaging techniques has provided many promising chemical leads against these three diseases [126], thereby giving rise to new drugs entering clinical trials against trypanosomatids (Figure 4). 

Among compounds with clear antiparasitic potential, a new class of cyclic boron-containing agents called benzoxaboroles has been introduced against other infectious diseases, such as toenail onychomycosis and atopic dermatitis [127]. The in vivo efficacy of 6-carboxamide benzoxaboroles was tested against HAT, Chagas disease, VL and CL. A phase 2/3 trial in patients of both stage 1 and stage 2 *gambiense*-HAT in the Democratic Republic of the Congo and Guinea started late in 2016 with acoziborole (Figure 4a) after a good-tolerance trial concluded in 2015 in France [128]. The study concluded in 2020, and the promising results have continued for pedriatic and *gambiense*-HAT seropositive non-parasitologically confirmed research participants (https://dndi.org/research-development/portfolio/acoziborole/; accessed on 3 March 2023). The benzoxaborole DNDI-6148 (Figure 4b) has been recognized as a promising lead against both Chagas disease and VL. The oral administration of DNDI-6148 was as curative as miltefosine in mice at a minimum dose of 25 mg/kg/day bid after a 10-day treatment, without toxic side effects [129,130,131]. This remarkable efficacy is currently being evaluated for first-in-human trials of safety, tolerability and pharmacokinetics after a single ascending oral dose (https://dndi.org/work-with-us/2022/rfp/phase-1-study-of-safety-tolerability-pharmacokinetics-dndi-6148/; accessed on 3 March 2023). Recently, the benzoxaborole prodrug AN15368 has been reported to be effective in the treatment of *T. cruzi* infections in non-human primates. In addition, this compound is orally active and exhibited no overt toxicity in a 60-day course, revealing it as a strong candidate for future clinical trials against Chagas disease [132].

The pyrazolopyrimidine GSK3186899 (Figure 4c) was first identified in a target-based screen against *T. brucei* glycogen synthase kinase (GSK3) [133]. Later, this family of compounds proved to be strong inhibitors of leishmanial cyclin-dependent kinase 12 (CRK12) with potent antileishmanial effects in vitro [134]. This compound, administered orally at 25 mg/kg bid for 10 consecutive days, produced a 99% reduction in parasite load in a mouse model of VL [135].

Proteasome inhibition has been identified by several groups as an interesting druggable target for the design and development of new molecules for the treatment of leishmaniasis, HAT and Chagas disease [136]. On the one hand, Novartis, in collaboration with the Wellcome Trust, identified in a first target-based screen an azabenzoxazole called GNF6702 (Figure 4d) as a potent reversible inhibitor of the kinetoplastid proteasome at submicromolar concentrations [136]. A first-in-class GNF6702 derivative called LXE408, administered at 3 mg/kg bid, showed a >99% reduction in the parasite liver load of *L. donovani* [137]. Simultaneously, GSK and Dundee Drug Discovery identified other leishmania proteasome inhibitors, thus demonstrating that the orally administered compound DDD01305143 (Figure 4e) reduced parasite burden by more than 95% in vivo in rodents at a dose of 25 mg/kg bid for 10 consecutive days [138]. Both of the proteasome inhibitors LXE408 and DDD01305143 have been promoted by DNDi to phase 1 clinical trials against VL (https://dndi.org/wp-content/uploads/2021/12/DNDi-RD-Portfolio-December-2021; accessed on 3 March 2023). Finally, CpG-D35 oligonucleotide, a synthetic short-chain immunomodulator, has been shown to be effective against CL in macaques. This compound, administered parenterally, reduced parasite load and lesion size through a systemic and long-lasting immunomodulatory effect that accelerates animal healing and produces long-lasting protective responses [139,140].

## 4. Development of Nitroheterocyclic Drugs against Trypanosomes

Nitroaromatic rings are privileged pharmacophores of many antimicrobials in clinical use, including antifungal and antiparasitic drugs. This definition includes a variety of molecules with a nitro functional group attached directly to an aromatic system ring (benzene), heterocycle (imidazole, triazole, thioimidazole, furan and others) or bicycle system [141]. Despite the multiple therapeutic uses of these molecules, the presence of a nitro group bound to electron-withdrawal moieties was considered an unwanted functionality in the hit optimization phase because it may cause toxicological issues, including hepatotoxicity, mutagenicity, carcinogenicity, etc. [142]. However, the urgent need for new candidates against NTDs and the use for more than 40 years of benznidazole and nifurtimox against sleeping sickness and American trypanosomiasis, the recent incorporation of fexinidazole, and the use of preventive and/or palliative treatments to face the side effects produced by nitro-containing drugs have reopened the field of research on these molecules. Some of them comply with the recommended Target Product Profile (TPP) for these diseases and are in an advanced state of clinical development. In addition, the identification of nitroaromatic compounds in drug discovery campaigns for repositioning drugs against trypanosomatids has created further interest in these compounds [143]. Detailed information about the nitro derivatives synthesized so far with antitrypanosomal activity has been published recently [144]. 

### 4.1. Nitroimidazoles

Nitroimidazoles are broad-spectrum antimicrobial drugs that have activity against anaerobic Gram-positive and Gram-negative bacteria, parasites and mycobacteria, and are used for the treatment of *Helicobacter pylori* and *Mycobacterium tuberculosis* infections [145]. Since new active drugs that effectively shorten the duration of treatment are urgently required to develop effective chemotherapies to combat NTDs, drug repurposing of nitroimidazoles represents a promising source of antitrypanosomal and antileishmanial agents.

#### 4.1.1. Fexinidazole

The 5-nitroimidazole derivative fexinidazole (Figure 2e) is also being developed as a potential treatment for other NTDs, as inferred by the ongoing clinical trials supported by the Drugs for Neglected Diseases initiative (DNDi) [67]. 

Fexinidazole undergoes rapid hepatic metabolization to sulfoxy and sulfone metabolites, which retain in vitro activity against *T. cruzi* in the low micromolar range. Fexinidazole has been reported to promote massive disorganization of reservosomes, the detachment of the plasma membrane, unpacking of nuclear heterochromatin, mitochondrial swelling, Golgi disruption and alterations in the kinetoplast-mitochondrion complex in *T. cruzi* [146]. In vivo efficacy after oral administration in a murine model of Chagas disease was confirmed at 50 mg/kg/day (30–40% cure rate) and 100 mg/kg/day (100% cure rate) [147,148]. The antichagasic effect of the final sulfone metabolite was experimentally confirmed in vivo in a chronic infection of *T. cruzi*, which could be cured with oral fexinidazole sulfone at 100 mg/kg/day for 5 days [149]. Due to the very promising results in preclinical studies with this drug, a phase 2 study of fexinidazole was initiated in 2014 in Cochabamba and Tarija, Bolivia (NCT02498782). Although promising results were obtained, the study had to be interrupted due to safety and tolerability issues. Another phase 2 proof-of-concept study using shorter and lower-dose treatment regimens in adults with chronic Chagas disease (NCT03587766) began in October 2017 in Spain and was completed with final analyses not yet concluded. Results showed that the efficacy of fexinidazole at lower doses and durations was not confirmed after 12 months. Potential use in combination with other drugs or for new indications, such as for immunocompromised patients at risk of reactivation, is being reviewed for the future development of fexinidazole as an antichagasic drug (https://dndi.org/research-development/portfolio/fexinidazole-chagas/; accessed on 3 March 2023). 

Fexinidazole and its metabolites were also tested against intracellular amastigotes of *L. donovani*, yielding a potency similar to miltefosine for the sulfoxy and sulfone forms, but little activity for the parent drug. Assessment of efficacy of fexinidazole in a mouse model of VL revealed that five single daily doses of 200 mg/kg were able to suppress the infection by 98.4%, which was equivalent to that reported for the clinical drugs in use [150]. New World *Leishmania* species were also susceptible to fexinidazole in vitro and in vivo. Hamster models of CL caused by *L. braziliensis* and *L. guyanensis* developed to test the efficacy of fexinidazole showed that this compound became effective from a dose of 200 mg/kg/day, thus resulting in a significant reduction in lesion size. Similarly, a mouse model developed to test the efficacy of oral fexinidazole against *L. amazonensis* showed a significant effect at 300 mg/kg/day [151]. A phase 2 proof-of-concept trial (NCT01980199) to determine efficacy of fexinidazole at a daily dose of 1800 mg (3 tablets) once a day for 4 days continued by 1200 mg (two tablets) once a day for 6 days in VL patients in Sudan was carried out by DNDi. All patients showed clinical improvement during treatment, and the majority had parasite clearance at the end of treatment. Three patients remained cured until 6 months follow-up. However, the response was not sustained in other patients and relapses were observed. The study was interrupted in 2014 as it failed to show conclusive efficacy in the majority of patients (https://dndi.org/research-development/portfolio/fexinidazole-vl/; accessed on 3 March 2023).

In addition to fexinidazole, there are other nitroimidazole compounds in different stages of development against trypanosomatids (Figure 5).

#### 4.1.2. Metronidazole Derivatives 

Metronidazole (2-methyl-5-nitroimidazole-1-ethanol) (Figure 5a) was used on patients with Chagas disease, but the results were very inconclusive [152,153]. However, in search of new strategies to reduce the benznidazole administration regimen in Chagas disease, combination therapies with metronidazole were studied. The combination of benznidazole:metronidazole in fixed ratios led to an increase in the potency of benznidazole in vitro. Preclinical studies in infected mice showed that when benznidazole (10 mg/kg) and metronidazole (250 mg/kg) were combined cumulative mortality was less than 40%. Although this combinatorial approach could not reduce the entire parasite load, it protected against cardiac issues induced by experimental *T. cruzi* infection [154].

Metronidazole has been also administered intralesionally to patients suffering CL as a 0.5% solution for the local treatment of small, non-inflamed and localized lesions. With a success rate of 87%, this treatment was safe, with no serious side effects [155]. However, intralesional injection of 10 mg metronidazole was not as effective as intralesional injection of Pentostam in the treatment of CL in Iraq [156]. Similar results were obtained in a clinical study conducted in Sri Lanka in 2019 with a group of patients with *L. donovani* CL. Although Pentostam was more effective in healing *Leishmania* lesions, metronidazole was the most cost-effective drug, curing more than 75% of patients [157].

Quinoline-metronidazole hybrid compounds have been synthesized and tested for antileishmanial activity. Two of these compounds, 15b and 15i (Figure 5b), showed activity against *L. donovani* promastigotes and amastigotes (>93% parasite killing) at 50 μM, with IC_50_s = 9.54 and 5.42 μM, respectively, against extracellular promastigotes, and IC_50_s = 9.81 and 3.75 μM, respectively, against intracellular amastigotes, with no cytotoxicity toward host mammal cells. In vivo studies were carried out with a mouse model of *L. donovani* infection with these compounds, which were administered intraperitoneally for 5 consecutive days in a dose range of 12.5–200 mg/kg daily. The results showed that both compounds were able to inhibit parasitic load in the organs of infected mice, with compound 15i exhibiting better efficacy than 15b, and inhibit parasite burden in the livers and spleens (>80%) of infected mice treated with a dose of 50 mg/kg/day for 5 consecutive days [158].

#### 4.1.3. Nitroimidazo-Oxazines and Nitroimidazo-Oxazoles

The 4-nitroimidazo-oxazine pretomanid (PA-824), (S)-2-nitro-6-[4-(trifluoromethoxy) benzyloxy]-6,7-dihydro-5H-imidazo [2,1-b][1,3]oxazine) (Figure 5c), which was approved in 2019 by the FDA to treat highly challenging cases of tuberculosis [159], also exhibited antiparasitic activity against trypanosomes and leishmanias. PA-824 consists of a mix of stereoisomers R and S with different biological activities in vitro. The R enantiomer was shown to be five times more active than the S enantiomer against all forms of the parasite. Treatment with 100 mg/kg with the R-stereoisomer of PA-824 suppressed infection by 99.9% and effectively cured the murine model of *L. donovani* VL in contrast to the S-stereoisomer, which was no more than 35% effective with the same dosification [160]. These results were also replicated in vitro against mammalian stages of *T. brucei* and *T. cruzi*, where the S-stereoisomer was largely inactive, in contrast to the R-stereoisomer, which showed higher trypanocidal activity [160]. 

PA-824 has served as a chemical scaffold for many other compounds with antitrypanosomal activity. From a library of ∼900 pretomanid analogs, compound DNDI-8219 ((6R)-2-Nitro-6-[4-(trifluoromethoxy)phenoxy]-6,7-dihydro-5H-imidazo [2,1-b][1,3]oxazine)) (Figure 5d) was identified. This compound showed strong (submicromolar) antileishmanial in vitro activity against both *L. infantum* and *L. donovani* intramacrophage amastigotes, with broad-spectrum activity against a range of reference strains and clinical isolates of *Leishmania*. In a mouse model of chronic VL, this compound gave rise to ≥97% inhibition when administered at 25 mg/kg bid for 5 days [161,162]. DNDI-8219 has shown interesting antichagasic activity against a chronic infection of *T. cruzi* mouse model when administered at 50 mg/kg/day for 5 consecutive days [163].

Delamanid (OPC-67683) is a nitro-dihydro-imidazooxazole (Figure 5e) resembling PA-824 that received conditional approval by the EMA for the treatment of multidrug-resistant tuberculosis [164]. Like PA-824, delamanid is a racemic mixture of two steroisomers, the R enantiomer showing better in vitro activity (submicromolar order) against promastigotes, axenic amastigotes and intracellular amastigotes of *L. donovani* than the S enantiomer. Both enantiomers provided good safety results in mammalian cells [165]. The same study tested the in vivo efficacy of an oral formulation of delamanid at different doses in a murine model of *L. donovani*, showing that either 30 or 50 mg/kg/day orally administered twice daily for 5 consecutive days compared favorably with the standard miltefosine regimen [165].

From the in vitro evaluation of a library of 72 nitroimidazo-oxazoles against *L. donovani* intracellular amastigotes, 25 compounds with IC_50_ values ≤0.1 μM were identified. Among them, the racemate DNDI-VL-2001 (2-methyl-6-nitro-2-[[4-(trifluoromethoxy)phenoxy] methyl]-2,3dihydroimidazo [2,1-b]oxazole), a 6-nitroimidazo-oxazole derivative, was able to reduce by 99% the parasite load in the liver of a mouse model of *L. donovani* VL at a dose of 25 mg/kg/day for 5 consecutive days via the oral route. R and S enantiomers were tested orally in a mouse model with five dose regimens ranging from 0.78 to 12.5 mg/kg. The R enantiomer DNDI-VL-2098 (Figure 5f) was identified as the most potent and safe stereoisomer. In addition, in a hamster model of VL, DNDI-VL-2098 promoted T-cell differentiation towards a Th1 type response and induced the production of extracellular nitric oxide [166]. Further QSAR studies of this compound were conducted in order to improve aqueous solubility and other drug-like features without compromising in vivo efficacy [167]. However, DNDI-VL-2098 did not continue to further studies due to safety issues (https://dndi.org/research-development/portfolio/vl-2098/; accessed on 3 March 2023).

The 7-substituted nitroimidazooxazine analogue of DNDI-VL-2098, DNDI-0690 (Figure 5g) was also identified from a library of 70 nitroimidazole-derived compounds by DNDi through an agreement with the TB Alliance. In mice, DNDI-0690 is rapidly absorbed into the bloodstream, showing a gastrointestinal absorption delay of 2.5 h before reaching peak plasma concentration and an estimated elimination half-life of >4 h [168]. The strong antileishmanial effect of this compound was demonstrated against intramacrophage amastigotes of *L. donovani* and *L. infantum* in vitro and in a hamster model of *L. infantum* VL in vivo, where it produced >99% inhibition of parasite load when administered orally at a dose of 12.5 mg/kg bid for 5 days [169]. In addition to its activity against VL, DNDI-0690 demonstrated excellent in vitro activity against three Old World and three New World strains of cutaneous *Leishmania* (EC_50_ < 5 μM). In a mouse model of *L. major* CL, DNDI-0690 exhibited > 95% efficacy at a dose of 50 mg/kg, while topical solutions applied directly to the skin lesion were < 50% active [129]. In another study, when DNDI-0690 was administered orally at a dose of 50 mg/kg daily for 10 consecutive days to two mouse models of *L. major* and *L. mexicana* CL in vivo, complete parasitological cure occurred [168]. Film-forming systems for delivery have been developed to increase the effectiveness of DNDI-0690 for the topical treatment of CL. In a *L. major* mouse model of CL, only the Eudragit RS formulation (containing butyl sebacate) resulted in a reduced parasite load, but without reduction in lesion size, which suggested that inadequate amounts of DNDI-0690 reached the parasites in the dermal layers of the skin upon topical application [170]. A multiple ascending dose phase 1 study of DNDI-0690 run in early 2020 in healthy adults demonstrated a favorable safety profile for single-dose administration (NCT03929016).

Recently, a second generation of DNDi-0690 analogues with increased solubility and safety led to the pyridine derivate (7-Methyl-2-nitro-7-({[5-(trifluoromethyl)pyridin-2-yl]oxy}methyl)-6,7-dihydro-5H-imidazo [2,1-b][1,3]oxazine) (Figure 5h). It exhibited increased bioavailability in mice and similar efficacy (reduction in the parasite burden >96%) in a model of VL when administered orally at a dose of 50 mg/kg bid for five days to mice infected with *L. donovani* [171].

### 4.2. Nitrofurans

Nitrofurans (Schiff base derivatives of 5-nitrofuraldehyde) are broad-spectrum redox-active antibiotics with dose-dependent bacteriostatic or bactericidal activity [172,173]. These compounds have been used in animal feeds, pharmaceuticals and other applications [174]. The hydrazine moiety of these compounds aids in the overall chemical stability of the nitrofuran ring due to its zwitterionic properties [175,176] and is responsible for the antipathogenic activity of nitrofurans [174].

There are several nitrofuran compounds in different stages of development against trypanosomatids (Figure 6).

#### 4.2.1. Nitrofurantoin

Nitrofurantoin (1-[[[5-nitro-2-furanyl]methylene]amino]-2,4-imidazolidinedione) (Figure 6a) is used for the treatment of human bacterial urinary tract infections [177] and has been listed in the WHO model list of essential medicines [87]. Early studies reported the in vitro antiparasitic activity of nitrofurantoin against certain strains of *T. cruzi* [178]. Nitrofurantoin and other N-alkyl and benzyl-substituted analogs were recently tested for their trypanocidal activity, showing strong trypanocidal activity (submicromolar range) against *T. congolense* and *T. b. rhodesiense* and low cytotoxicity. However, only nitrofurantoin had significant trypanocidal activity when it was administered orally at a dose of 10 mg/kg daily for 7 consecutive days to mice infected with *T. congolense* IL3000 strain [179]. In an ascending dose experiment with nitrofurantoin administered orally to mice infected with *T. congolense*, doses of 30 to 100 mg/kg for 7 consecutive days were safe and curative and resulted in the complete healing of the animals [180], thus pointing to nitrofurantoin as a potential scaffold for the synthesis of new compounds with antitrypanosomal activity.

#### 4.2.2. Nitrofurazone 

Nitrofurazone (5-nitro-2-furaldehyde-semicarbazone), a hydrazinecarboxamide containing 5-nitrofuran, lacks the second cyclic substituent in its structure (Figure 6b). Formulated as a topical cream, it is used for the treatment of superficial wounds, second- and third-degree burns, ulcers and skin infections [181,182]. The trypanocidal activity of nitrofurazone was earlier demonstrated on the intracellular forms of *T. cruzi* in experimental Chagas disease [183]. More recently, a novel nitrofurazone derivative, hydroxymethylnitrofurazone, together with nitrofurazone, was tested in cell cultures (LLC-MK2) infected with amastigote and trypomastigote forms of *T. cruzi*. The nitrofurazone derivative exhibited better trypanocide activity and lower mutagenicity potential than nitrofurazone, with 100% inhibitory activity against amastigotes and trypomastigotes at 5 μM [184].

#### 4.2.3. Other Nitrofuran Derivatives

It has been reported that several 5-nitro-2-furancarbohydrazines (Figure 6c) demonstrated good activity against the intracellular amastigote stage of both *T. cruzi* and *L. infantum* and *T. brucei* bloodstream trypomastigotes [185]. Some compounds derived from palladium nitrofurylthiosemicarbazone ([PdCl_2_(HL)] and [Pd(L)_2_]) (Figure 6d), which are trypanothione reductase inhibitors, were 1.7-fold more active than nifurtimox against *T. cruzi* epimastigotes [186]. 

The melamine derivate of nitrofuran WSP934 (Figure 6e) was reported to have very potent activity against *T. b. rhodesiense* (IC_50_ about 11 nM). In addition, when 20 mg/kg of this compound was administered daily by the intraperitoneal route for 4 consecutive days to mice infected with *T. b. brucei*, a curative effect was observed [187,188]. In another study [189], when the dose was increased to 40 mg/kg/day, mice infected with *T. b. rhodesiense* were also cured. These authors also observed that oral treatment with 100 mg/kg for 4 days gave rise to moderate activity, whereas 10 doses of 50 mg/kg given intraperitoneally did not provide cure in a stage 2 rodent model of infection.

5-Nitrofuryl derivatives containing thiosemicarbazones (Figure 6f) have been reported to possess high anti-*T. cruzi* activity. In vitro activity against *T. cruzi* of some of these molecules (6, 8, 10 and 12), which are 3-(5-nitrofuryl) acroleinyl derivatives containing H, methyl, ethyl and phenyl moieties, respectively, was superior to that of nifurtimox [190]. In another study, 19 compounds containing a common 5-nitrofuran core with different substituent groups at the 2-position on the furan ring (thiosemicarbazones, semicarbazones and carbazates) were tested in vitro for their activity against the bloodstream-form of *T. brucei*. From the 19 compounds, 13 had IC_50_s <1 μM, with the thiosemicarbazones HC1, HC2 and HC4, the carbazates HC10 and HC11, and the semicarbazone nitrofurazone providing values of <250 nM. The carbazate HC10 was the most potent agent against both bloodstream forms of *T. brucei* and mammalian cells, with a promising selectivity index of 116 [191].

A series of nitroheterocyclic compounds (BSF series) were tested as new alternatives against *L. infantum* [192]. Five compounds, 4-R-substituted-N′-[(5-nitrofuran-2-yl) methylene] benzhydrazide (R = H, Cl, NO_2_, CH_3_, C_4_H_9_) (Figure 6g), were obtained through the molecular modification of nifuroxazide, 4-hydroxy-N′-[(5-nitrofuran-2-yl)methylene] benzohydrazide and provided activity against this parasite in a dose-dependent fashion. BSF-H, BSF-Cl and BSFNO_2_ provided the best IC_50_ values in promastigotes (0.76 μM, 0.72 μM and 0.58 μM, respectively). The authors also showed that BSF-H and BSF-Cl were effective in infected macrophages, too, thereby concluding that these nitroheterocyclic compounds were promising antileishmanial drugs.

In a recent published screening of two commercial collections of 1,769 repositioning drugs using splenic explants from mice infected with an infrared fluorescent *L. donovani* strain [193], 43 compounds with leishmanicidal activity > 1μM were selected. Among the selected compounds, several with heterocyclic structures were identified as the most potent, including nifurtimox, nitrofurantoin, nitrofurazone, PA-824, fexinidazole and the fungicide Nifuratel (Macmiror) (Figure 6h), the latter being a synthetic nitrofurantoin derivate active against most agents causing genital urinary infections [194], whose potency and selectivity indicate it as a promising antileishmanial drug [143].

Further studies in a mouse model of chronic VL with a dose of nifuratel administered orally at 50 mg/kg bid for 10 consecutive days resulted in a reduction in parasite load of more than 80%. In addition, intralesional administration of nifuratel in a model of *L. major* CL resulted in parasitological cure, making this compound a potential candidate for the treatment of this insidious form of the disease [143]. Nifuratel was also tested in combination with miltefosine, and the 1/30 ratio (nifuratel/miltefosine) exhibited synergic antileishmanial effect both in vitro in axenic amastigotes of *L. donovani* and ex vivo in intramacrophagic forms. Administration of 50 mg/kg nifuratel (twice daily) in combination with 10 mg/kg miltefosine (once daily) for 10 days produced in a mouse model of VL a significant reduction in total parasite burden, especially in the thymus, liver and bone marrow, although it was not able to completely clear the presence of parasites in the spleen [195].

### 4.3. Other Nitroheterocycles

Other active nitroheterocyclic compounds have been tested against trypanosomatids (Figure 7).

From an in-house library comprising 76 nitroheterocycles and related compounds tested in vitro against *T. b. rhodesiense*, six hits were identified showing interesting activity (IC_50_ ≤ 10 μM) and fair selectivity (SI > 17). The best antitrypanosomal compounds were those with the quinoxalin-2-one (Figure 7a) scaffold (IC_50_ values <15 μM and SI values in the range of 3.6–39.7). Two compounds, which derived from the 7-nitroquinoxalin-2-one and 5-nitroindazole scaffolds, respectively, were found to be of particular interest because of their established oral bioavailability in mice [196].

A library of nitrobenzylphosphoramide mustards (Figure 7b) was assessed in vitro against *L. major*. Five compounds showed significant activities (IC_50_ values of <10 μM) against both parasite forms of *L. major*, showing no toxicity to the mammalian cells at concentrations up to 100 μM. Three of these compounds contained halogen substituents on the nitrobenzyl ring and were identified as being preferred substrates for the *L. major* type I NTR and provided IC_50_ values ≤ 1.09 μM against the intracellular stage and SIs > 92 [197].

3-Nitro-1H-1,2,4-triazole- and 2-nitro-1H-imidazole-based amides with aryloxy-phenyl cores (Figure 7c) demonstrated significant activity against *T. cruzi* amastigotes. The 3-nitrotriazole-based derivatives showed high potency as anti-*T. cruzi* agents at subnanomolar concentrations with high selectivity. Antitrypanocidal activity of the 2-nitroimidazole analogs was only moderate and poorly selective. Interestingly, both types of compounds were also active against *L. donovani* axenic amastigotes, with excellent selectivity for the parasite. Despite the activity against *L. donovani* axenic amastigotes, 3-nitrotriazole-based analogs were not particularly active in infected macrophages, whereas three 2-nitroimidazole-based analogs were also moderately active against infected macrophages. On the contrary, these compounds could not demonstrate selective activity against *T. b. rhodesiense*. Mice infected with *T. cruzi* were able to reduce the parasite load to undetectable limits after a 5-day intraperitoneal treatment with 13 mg/kg/day of three 3-nitrotriazole-based aryloxyphenylamides. These compounds constitute a promising generation of antitrypanosomal agents [198].

Fifteen compounds based on a 2-amide 5-nitrothiazole structure (Figure 7d) were tested for growth-inhibitory activity against the bloodstream form of *T. brucei*. Only compounds NT2, NT4, NT6, NT7 and NT11 provided appreciable trypanocidal activity (IC_50_s < 10.0 μM), with no growth-inhibitory effects at concentrations of up to 100 μM in mammalian cells [199].

Several 3-nitroimidazo [1,2-α]pyridine derivatives (Figure 7e) were synthesized and evaluated for their in vitro antiparasitic activity in both *Leishmania* spp. and *T. b. brucei*. Although the biological activity of these compounds against *Leishmania* was poor, they inhibited bloodstream forms of *T. b. brucei* in the submicromolar range, with excellent selectivity index values (≥2500) in mammalian cell cultures. The best antitrypanosomal molecule in this series (compound 8: 8-bromo-6-chloro-2-(methylsulfonylmethyl)-3-nitroimidazo [1,2-a]pyridine) showed an EC_50_ = 17 nM and an SI = 2650, was not genotoxic, displayed improved aqueous solubility and better in vitro pharmacokinetic properties, and was well tolerated by mice after repeated oral administrations of 100 mg/kg for 5 days [200].

## 5. Activation of Nitroheterocyclic Compounds as a Key Factor in Their Use as Antitrypanosomatid Drugs

Given the importance of nitroheterocyclic compounds in the treatment of parasitic diseases, a major effort is being made to elucidate their mechanisms of action. However, the fact that nitroheterocyclic compounds can be enzymatically activated by a nitroreductase enzyme (NTR) allows them to kill the parasite by additional mechanisms [118,141,201].

Two classes of NTRs are involved in the enzymatic catalysis of nitro groups: the type I NTR (oxygen-insensitive) catalyzes two reduction reactions of the nitro group, whereas the type II NTR (oxygen-sensitive) catalyzes just one reduction reaction. The reaction performed by type I NTR results in the production of an aromatic amine via nitroso and hydroxylamine intermediates. Both the intermediates and the products generated in this reaction can interact with essential intracellular molecules and cause cellular damage [202,203]. The reduction by type I NTR does not generate reactive oxygen species, as oxygen is not involved as a cosubstrate. [197,204,205]. On the contrary, the reaction carried out by the type II NTR generates an unstable nitro radical anion, which is reoxidized back to a nitro group by molecular oxygen, which in turn is converted into a reactive superoxide anion [206]. Free radicals can readily react with cellular macromolecules, and as a result, lipid oxidation, cell membrane damage, enzyme inactivation and, finally, the fragmentation of DNA sequences is observed [207].

Initially, the oxidative stress caused by type II NTR activation was proposed as the main mechanism of action of nitroheterocyclic compounds. Nevertheless, subsequent experiments in which the overexpression of leishmanial and trypanosomal type I NTR increased the sensitivity of these parasites to nitroheterocyclic compounds between 9- and 19-fold [150,204,208,209,210,211] indicated that monocyclic nitro compounds, such as nifurtimox, benznidazole, fexinidazole and its sulfonic metabolite, are essentially activated by type I NTR [191,197,204]. On the other hand, the bicyclic nitro compounds delamanid and PA-824 are essentially activated by an uncommon NAD(P)H-dependent flavoprotein called NTR2 in *Leishmania* spp. [148,212], as was evidenced by experiments in which hypersensitivity to PA-824 (~40-fold more susceptible) was only observed after the overexpression of NTR2 and not that of type I NTR [160,212].

Although the natural function of NTR2 in *Leishmania* spp. is unknown, it has high similarity to prokaryotic alkene reductases from the so-called “old yellow enzyme” family. The members of this family catalyze a varied range of reactions, usually on substrates with an α/β unsaturated carbonyl group [212,213,214,215]. Regarding other trypanosomatids, BLAST analysis showed that the enzyme prostaglandin F2α synthase, also known as “old yellow enzyme”, shares 44% identity with the NTR2 of *L. donovani*. This enzyme has been related to the mechanism of action and resistance of the Chagas disease treatments benznidazole and nifurtimox [216,217]. However, an enzyme equivalent to NTR2 was not found in *T. brucei*, thereby giving a possible explanation for the lack of effect of the bicyclic nitro compounds in this parasite [160,212].

The mechanism of action by which the intermediates and products generated in the activation of nitroheterocyclic compounds by type I NTR cause cell death is unknown. Until 2021, several hypotheses about the mode of action have been suggested. For example, it has been hypothesized that the unsaturated open-chain nitrile produced in the metabolism of nifurtimox could inhibit essential proteins by covalent binding to cysteine residues, since this unsaturated open-chain nitrile contains a Michael acceptor, which is a chemical moiety that interacts with the sulfhydryl groups in biomolecules [141,203]. However, in 2021, Dattani and co-workers provided the first direct link between the activation of benznidazole and its trypanocidal activity, thus demonstrating that benznidazole produces DNA damage in the nuclear genome of *T. brucei* in a type I NTR-dependent process and pointing out that it most likely induces double-strand DNA breaks [218].

Trypanosomal type I NTR not only plays a key role in the antiparasitic activity of nitroheterocyclic compounds, but also is responsible for its antitrypanosomal selectivity, since it is absent from mammals and is found in bacteria, along with some fungi and protozoans. For instance, whereas the unsaturated open-chain nitrile generated in the metabolism of nifurtimox is equally potent in *T. brucei* and mammalian cells in vitro (EC_50_ values of 5.3 μM and 2.9 μM, respectively) [118], nifurtimox alone is 10-fold more potent in *T. brucei* than in the host, thus confirming this paradigm.

The restricted distribution of this class of enzymes also explains why nitroheterocyclic compounds, such as fexinidazole, are genotoxic in the standard Ames test but do not show any toxicity against mammalian cells in in vitro assays. The genotoxicity observed in the Ames test is due to activation by bacterial type I NTR (similar to the enzymes found in trypanosomes)—a reaction that cannot be conducted by the mammalian type II NTR. Therefore, nitroheterocyclic compounds should be deeply examined, because they could be functional and non-genotoxic [189,191,204].

Inhibition of trypanothione reductase has also been suggested as a mechanism of action according to molecular modeling studies carried out with 7-nitroquinoxalin-2-one derivatives [219]. This enzyme has also been suggested to be the target of nitrofurans, acting in a different way in aerobic or anaerobic conditions. In the presence of O_2_, these redox-cycling substrates behave as subversive substrates for trypanothione reductase, not inactivating the enzyme but effectively inhibiting the enzymatic reduction of trypanothion, causing the production of free radicals and leading to futile consumption of NADPH, which can contribute to cytotoxicity. Under anaerobiosis, the nitrofuran compounds cause irreversible inactivation of the enzyme [220]. Inhibition of this enzyme following an uncompetitive enzyme inhibition pattern has been reported with new synthetic compound derivatives from 5-nitro-2-furoic acid [205] and also with nifuratel [195]. In the case of the nitroindazoles 3-benzyloxy-1-methyl-5-nitro-1H-indazole and N-Methyl-5-(3-benzyloxy-5-nitro-1H-indazol-1-yl)-3-oxapentylamine, it has been proposed that their mechanism of action could be related to interference with some glycosomal or mitochondrial enzymes involved in the catabolism of *T. cruzi* [221].

Finally, the development of multiple nitroheterocyclic compounds at a time for trypanosomatid diseases could have complications on account of the high dependency on prodrug activation by a specific enzyme, given that this can also be a mechanism by which clinical resistance to nitroheterocycle compounds can emerge. In fact, not only cross-resistant parasites to two nitroheterocyclic compounds, nifurtimox and fexinidazole, have been described, but also the resistance mechanisms found have been related directly or indirectly to activation by NTR [119,222]. However, the discovery of nitroheterocyclic compounds activated in an independent type I NTR manner (bicyclic nitro compounds), along with the fact that any alteration of type I NTR is incompatible with the viability of *Leishmania* and *Trypanosoma* parasite replicating forms or severely compromises their ability to spread, mitigates many of these concerns [197,204,212]. As observed in the laboratory, the maximum level of fexinidazole resistance achieved in *Leishmania* promastigotes knockouts was two-fold more than in control cells [204]. Ultimately, it is fundamental to determine the bioactivation mechanism of the new nitroheterocyclic compounds in an early stage of its development.

## 6. Concluding Remarks

Compounds with nitroheterocyclic structures have been successfully used for more than 40 years as oral drugs against different pathogens, including those causing sleeping sickness and Chagas disease. These compounds were a source of inspiration for medicinal chemists to synthesize new molecules with better pharmacological profiles, but the potential toxicity phenomena due to the nitro group which is necessary for their biological activity recommended their limited use and even prevented their inclusion in drug discovery campaigns. Indeed, the electron-withdrawing properties of the nitro group, as well as the known bioactivation by different enzymatic systems, limited the development of new molecules, especially because of their predictable mutagenicity. However, the urgent need for active molecules against these diseases and the recent successes of fexinidazole and the antitubercular derivatives PA-824 and delamanid have led to renewed interest in these compounds among organic chemists, who, once again, have incorporated them into the designs of new synthetic drugs. It is equally striking that in recent blind screenings of repositioning drugs, compounds of nitroaromatic structure have shown great potential both in efficacy and selectivity, thereby demonstrating that they are not yet discardable pharmacophores for pharmacological development.

Among the latest additions of nitroaromatic drugs to the therapeutic portfolio for these diseases, it is worth mentioning the recent entry onto the market of fexinidazole for the treatment of HAT and its possible incorporation into the treatment of Chagas disease and the antileishmanial success of the nitroimidazooxazine DNDi-0690—a compound of great interest that is entering phase 2 clinical trials and showing strong effect against VL, with poor toxicity (including fetotoxic issues derived from the nitro function). Other derivatives of DNDi-0690 are waiting to fill the pipeline of active compounds in the event the drug candidate fails in some of the clinical phases in which it is being tested. In addition, the nitrofuran derivative Nifuratel, which was used against vaginal infections for decades, has been repositioned as a potent antileishmanial drug in a phenotypic screening against *L. donovani*. In vivo tests demonstrated its potency in a murine model of visceral leishmaniasis, with greater efficacy than glucantime administered intralesionally against *L. major* CL, where it produced parasitological cure. These results could be extended to other pathologies caused by trypanosomatids.

However, despite the enthusiasm produced by the results for these compounds due to their good TPPs against the three diseases considered, caution should be taken in their design and use in order to alleviate or reduce the side effects of these compounds. This will prevent as much as possible the risks associated with this interesting group of chemical compounds.

## Figures and Tables

**Figure 1 biomolecules-13-00637-f001:**
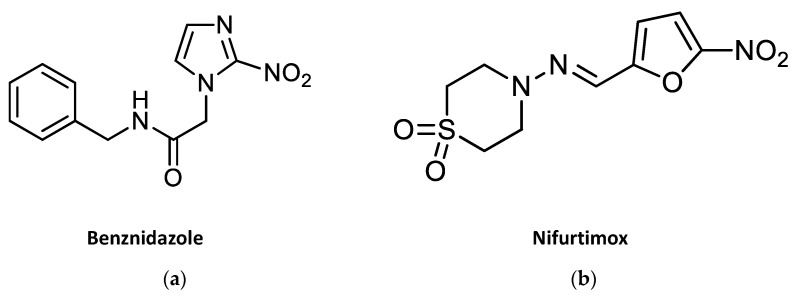
Drugs in clinical use against American trypanosomiasis (Chagas disease). (**a**) Benznidazole. (**b**) Nifurtimox.

**Figure 2 biomolecules-13-00637-f002:**
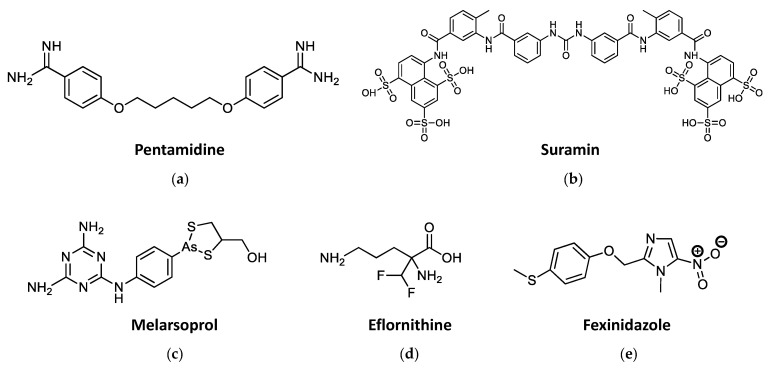
Drugs in clinical use against Human African Trypanosomiasis (HAT). (**a**) Pentamidine. (**b**) Suramin. (**c**) Melarsoprol. (**d**) Eflornithine. (**e**) Fexinidazole.

**Figure 3 biomolecules-13-00637-f003:**
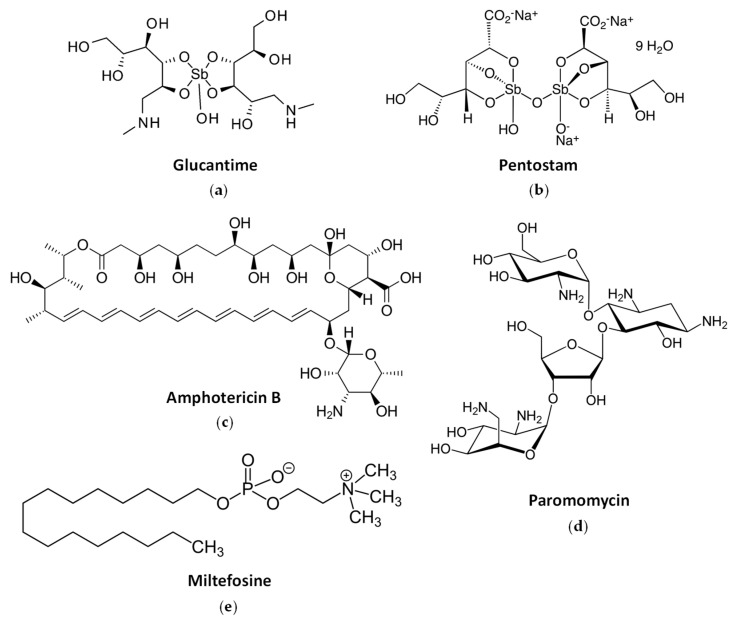
Drugs in clinical use against leishmaniasis. (**a**) Glucantime. (**b**) Pentostam. (**c**) Amphotericin B. (**d**) Paromomycin. (**e**) Miltefosine.

**Figure 4 biomolecules-13-00637-f004:**
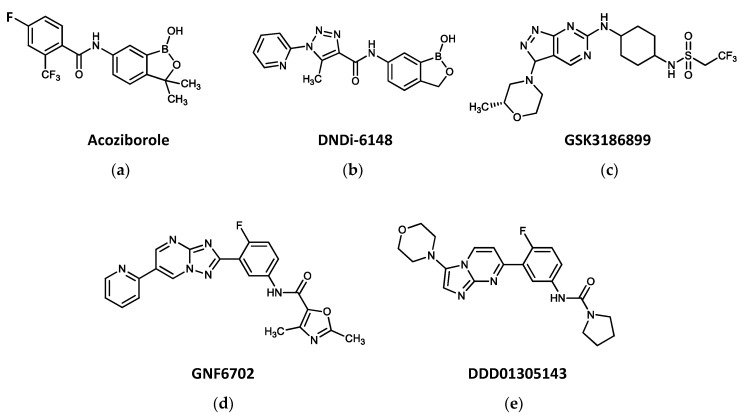
New drugs entering clinical trials against trypanosomatids. (**a**) Acoziborole. (**b**) DNDi-6148. (**c**) GSK3186899. (**d**) GNF6702. (**e**) DDD01305143.

**Figure 5 biomolecules-13-00637-f005:**
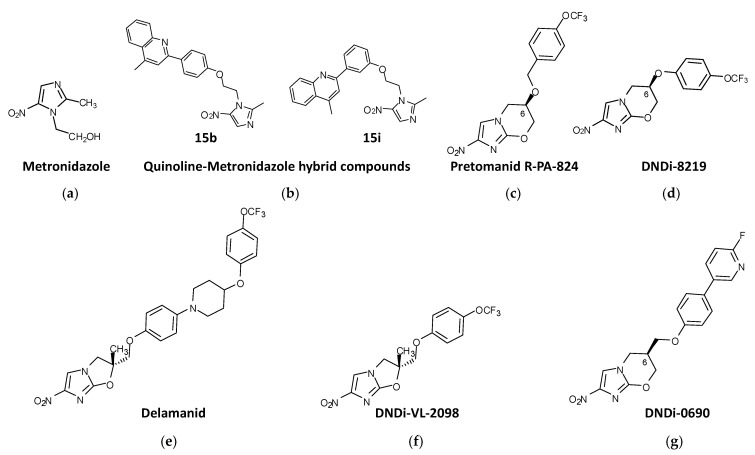
Nitroimidazole compounds in different stages of development against trypanosomatids. (**a**) Metronidazole. (**b**) Quinoline-Metronidazole hybrid compounds 15b and 15i. (**c**) Pretomanid R-PA-824. (**d**) DNDi-8219. (**e**) Delamanid. (**f**) DNDi-VL-2098. (**g**) DNDi-0690.

**Figure 6 biomolecules-13-00637-f006:**
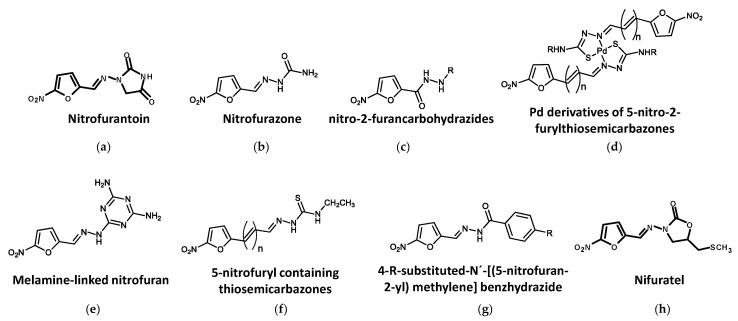
Nitrofuran compounds in different stages of development against trypanosomatids. (**a**) Nitrofurantoin. (**b**) Nitrofurazone. (**c**) 5-nitro-2-furancarbohydrazides. (**d**) Pd derivatives of 5-nitro-2-furylthiosemicarbazones. (**e**) Melamine-linked nitrofuran. (**f**) 5-nitrofuryl containing thiosemicarbazones. (**g**) 4-R-substituted-N′-[(5-nitrofuran-2-yl) methylene] benzhydrazide. (**h**) Nifuratel.

**Figure 7 biomolecules-13-00637-f007:**
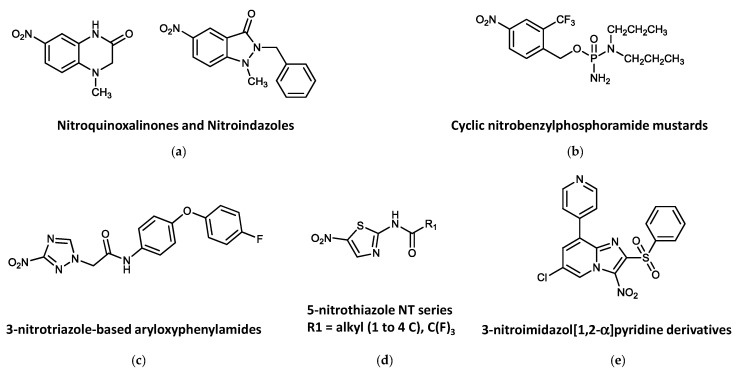
Other active nitroheterocyclic compounds tested against trypanosomatids. (**a**) Nitroquinoxalinones and Nitroindazoles. (**b**) Cyclic nitrobenzylphosphoramide mustards. (**c**) 3-nitrotriazole-based aryloxyphenylamides. (**d**) 5-nitrothiazole NT series. (**e**) 3-nitroimidazol [1,2-α] pyridine derivatives.

## Data Availability

Not applicable.

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
