# Peer review of "Further Investigations of Nitroheterocyclic Compounds as Potential Antikinetoplastid Drug Candidates"

_biomolecules, 2023, doi:10.3390/biom13040637_

Round 1

Reviewer 1 Report

Parasitic infections continue to be a problem world-wide and this review brings together a great deal of information relevant to existing and new treatments. To that extent, I have no doubt that it should be published and will be found useful by a broad scientific community. It is also appropriate to publish in the suggested journal issue. I should like to raise a number of points with the authors for consideration. 

1. I was misled by the title because a substantial part of the first 12 pages has nothing whatsoever to do with nitro compounds. Eventually, the point of the title becomes clear but far from 'inspiration', I would suggest that further investigations of nitro compounds are closer to 'me-too' drugs. I think that a different title is essential.

2. Holding the import of the title in mind, I wondered when reading first what the relevance of all of the clinical data for non-nitro drugs is, unless it is compared closely with that for the new nitro drugs, which I did not find it to be, although data are provided. The balance of the review needs reconsideration in my opinion.

3. Quite appropriately, drugs of many different families are mentioned to provide the context for advances in the development of nitro compounds. I would like to see a stronger scientific case built for further research into nitro drugs. Personally, I regard the newer compounds like fexnidazole as tweaks and repurposings of the old standards. This hardly suggests an inspirational place for nitro compounds in drug discovery.

4. I looked particularly for discussion of emerging resistance to existing drugs and found few comments until the very last pages. AMR is such an important aspect of modern anti-infective drug discovery that I would have expected it to merit a section of its own in a review like this. Such a section could also contribute to making a stronger case for nitro compounds.

5. Since the title of the journal contains the word 'molecules' I would have liked to have read more information about the molecular mechanism of action of nitro compounds. The same could be said about other classes of compound such as the oxyboroles but it's most important for the nitro compounds since they are the focus of the review. I suggest that there should be a more extended discussion of reduction of nitro groups in vivo to give the biologically active compounds. This is at the heart of selectivity, and explanations are implied but without adequate discussion and description.

6. It often appears that authors of papers related to medicinal chemistry like to include a molecular docking of one or two compounds with a relevant protein. This is the case here. I could not see what useful information in the general context of nitro compounds as drugs this study has. Two compounds and one enzyme do not make a design paradigm. If there's going to be molecular graphics used, in my opinion, they must show how the binding of the drug to the enzyme links with nitro group reduction, the key step of molecular mechanism.

7. Again, this is a personal reaction but if I were a pharmaceutical company executive, I would not be persuaded by this review of a case to invest in new nitro compounds. Repurposing is one thing, when the pharmacology is already well known, but the risks of investigating new inherently toxic compounds (nitro compounds) requires a stronger case. Perhaps on revising, the authors can make such a case.

Lastly, I emphasise that these are my personal reactions to reading the review. As I wrote at the top of this opinion, there is a lot of valuable information brought together. My concern is that its evaluation could be more helpful to readers.

Author Response

Response to Reviewer 1 Comments

Parasitic infections continue to be a problem world-wide and this review brings together a great deal of information relevant to existing and new treatments. To that extent, I have no doubt that it should be published and will be found useful by a broad scientific community. It is also appropriate to publish in the suggested journal issue. I should like to raise a number of points with the authors for consideration.

Point 1. I was misled by the title because a substantial part of the first 12 pages has nothing whatsoever to do with nitro compounds. Eventually, the point of the title becomes clear but far from 'inspiration', I would suggest that further investigations of nitro compounds are closer to 'me-too' drugs. I think that a different title is essential.

Response to 1: Thank you for the suggestion. Since the journal is not specific for parasitic diseases, first of all we tried to provide readers with a global overview about the neglected tropical diseases caused by trypanosomatids and the current treatments available, which include several nitro compounds, namely nifurtimox, benznidazole, fexinidazole and others entering in phase I clinical trials, which serve of justification of the importance of this family of compounds. Then, after this introductory background, we dive into the development of nitro compounds against these microorganisms.

The title has been modified in the revised MS, as suggested:

“Further investigations of nitroheterocyclic compounds as potential antikinetoplastid drug candidates”

Point 2. Holding the import of the title in mind, I wondered when reading first what the relevance of all of the clinical data for non-nitro drugs is, unless it is compared closely with that for the new nitro drugs, which I did not find it to be, although data are provided. The balance of the review needs reconsideration in my opinion.

Response to 2: Thank you for the suggestion. In our opinion, within the background we introduce indistinctly all the marketed drugs for these diseases including nitro drugs (nifurtimox, benznidazole, fexinidazole) and non-nitro drugs (DFMO, Sb- and As-based drugs, AmpB, and others). Nevertheless, in order to introduce the reader to this field of research, we decided to include the information regarding non-nitro drugs.

Point 3. Quite appropriately, drugs of many different families are mentioned to provide the context for advances in the development of nitro compounds. I would like to see a stronger scientific case built for further research into nitro drugs. Personally, I regard the newer compounds like fexnidazole as tweaks and repurposings of the old standards. This hardly suggests an inspirational place for nitro compounds in drug discovery.

Response to 3: Thank you for the comment. According to DNDi roadmap, one of the compounds entering in phase I clinical trials has nitroheterocyclic structure, and many medicinal chemistry groups are preparing new compounds with this structure. However, we have modified the title to delete the word inspirational.

Point 4. I looked particularly for discussion of emerging resistance to existing drugs and found few comments until the very last pages. AMR is such an important aspect of modern anti-infective drug discovery that I would have expected it to merit a section of its own in a review like this. Such a section could also contribute to making a stronger case for nitro compounds.

Response to 4: Thank you for the comment. A section for AMR has been added to the text. Lines 449-500 (revised MS)

“2.4. Drug resistance to current pharmacology

The lack of a varied offer of drugs against trypanosome diseases and their overuse is at the origin of their loss of efficacy and the appearance of resistances. Big pharmaceutical companies have little response to this situation due to the neglected nature of these diseases. However, it should not be forgotten that all parasites - which are in a constant arms race with the host - develop strategies to escape both the host's immune system and the drugs administered [108]. These resistances are sometimes generic, such as systems related to drug absorption and elimination, and sometimes specific to a particular drug. To these resistance processes, we must add the pathogen's ability to persist in dormant states - persisters - of low metabolism and lower sensitivity to drug treatment [109]. Finally, we must mention the lack of clear therapeutic targets in trypanosomatids, if any, which hinders and delays the discovery and development of new therapeutic entities against these diseases [110]. Although it is not the purpose of this review, we will briefly introduce some examples of drug resistance to trypanosomatids. Among the mechanisms of generic resistance to antileishmanial drugs, the case of SbV derivatives is well known.

For years Glucantime has been known to be therapeutically ineffective in the state of Bihar in Northeast India [111]. Resistance is associated with drug transport due to mu-tations in aquaglyceroporin 1 that have been linked to high levels of arsenic in drinking water [112]. Similarly, mutation or loss of aquaglyceroporin 2 is also implicated in the resistance of certain strains of T. b. gambiense to melarsoprol [113]. Resistance to miltefosine - easily reproducible in the laboratory by successive passes at increasing drug concen-trations - has been attributed to mechanisms of reduced drug absorption under field conditions [114].

In addition to the alteration of transport mechanisms, there have been cases of re-sistance linked to drugs with known mechanism of action, as is the case of eflornithine, the ODC inhibitor. In addition to mutations in the amino acid transporter AAT6 [115], T. b. brucei strains with silenced ODC and compensated with an upregulated putrescine transport mechanism were used to explain resistance to this drug, a component of NECT therapy [116,117]. In the case of nitro compounds such as nifurtimox (the second com-ponent of NECT therapy), resistant strains related to the prodrug nature of these com-pounds have also been described. Nifurtimox is a prodrug that must undergo two re-ductions in its nitro group catalyzed by type I NTRs, directly forming intermediate species toxic to the parasite [118]. Low levels of NTR I of trypanosomes directly affect their sen-sitivity to nifurtimox and other nitro derivatives, since reduced levels of the enzyme led to cross-resistance to other nitro compounds such as benznidazole and fexinidazole [119]. In the case of AmpB, cases of resistance are not very frequent and have been associated with mutations in sterol metabolism genes (sterol methyl transferase or the sterol C5 desaturase) [120].

Persistent phenotypes – a transient subpopulation that is less susceptible to drug treatment and may remain in a dormant state after drug treatment – have also been described in trypanosomatids. In Chagas disease, both the long-term persistence of parasites in patients and the frequent failure of standard therapies (benznidazole and nifurtimox) [121]. There is also evidence of persister forms in leishmaniasis. PKDL is an example of relapse occurring after treatment and apparent cure of VL with antimonials in India [122,123].

Evolution of drug resistance is a serious problem, since it makes the current treatments become less potent as they are administered over time. Therefore, new strategies are necessary to slow the rate of resistance to current treatments against diseases caused by trypanosomatids. One of these strategies is drug combination, which comprises the use of at least two drugs with different mechanisms of action. Those drugs must be carefully selected to minimize the acquisition of resistance in order to improve clinical outcome [124]”.

Point 5. Since the title of the journal contains the word 'molecules' I would have liked to have read more information about the molecular mechanism of action of nitro compounds. The same could be said about other classes of compound such as the oxyboroles but it's most important for the nitro compounds since they are the focus of the review. I suggest that there should be a more extended discussion of reduction of nitro groups in vivo to give the biologically active compounds. This is at the heart of selectivity, and explanations are implied but without adequate discussion and description.

Response to 5: Thank you for the comment. We have extended the discussion about the mechanism of action of nitro-derivatives.

Lines 911 to 913 (revised MS) “Free radicals can readily react with cellular macromolecules, and as a result, lipids oxidation, cell membrane damages, enzyme inactivation, and, finally fragmentation of the DNA sequence is observed [208]”. (Olender et al., 2018).

Lines 961 to 975 (revised MS) “Inhibition of trypanothione reductase has also been suggested as mechanism of action according to molecular modelling studies carried out with 7-nitroquinoxalin-2-one derivatives [220] (Aguilera-Venegas et al., 2011). This enzyme has also been suggested to be the target of nitrofurans, acting in a different way in aerobic or anaerobic conditions. In the presence of O2, these redox-cycling substrates behave as subversive substrates for trypanothione reductase, not inactivating the enzyme but effectively inhibiting enzymatic reduction of trypanothion, causing the production of free radicals, and leading to futile consumption of NADPH, which can contribute to cytotoxicity. Under anaerobiosis, the nitrofuran compounds cause irreversible inactivation of the enzyme [221] (Henderson et al., 1988). Inhibition of this enzyme following an uncompetitive enzyme inhibition pattern has been reported with new synthetic compounds derivatives from 5-nitro-2-furoic acid [206] (Arias et al., 2017) and also with nifuratel [195] (Melcón-Fernández et al., 2023). In the case of the nitroindazoles 3-Benzyloxy-1-methyl-5-nitro-1H-indazole and N-Methyl-5-(3-benzyloxy-5-nitro-1H-indazol-1-yl)-3-oxapentylamine, it has been pro-posed that their mechanism of action could be related to the interference with some glycosomal or mitochondrial enzymes involved in the catabolism of T. cruzi [222] (Muro et al., 2014).”

Point 6. It often appears that authors of papers related to medicinal chemistry like to include a molecular docking of one or two compounds with a relevant protein. This is the case here. I could not see what useful information in the general context of nitro compounds as drugs this study has. Two compounds and one enzyme do not make a design paradigm. If there's going to be molecular graphics used, in my opinion, they must show how the binding of the drug to the enzyme links with nitro group reduction, the key step of molecular mechanism.

Response to 6: Thank you for the comment. According to your suggestion we have deleted figure 8 in the revised version.

Point 7. Again, this is a personal reaction but if I were a pharmaceutical company executive, I would not be persuaded by this review of a case to invest in new nitro compounds. Repurposing is one thing, when the pharmacology is already well known, but the risks of investigating new inherently toxic compounds (nitro compounds) requires a stronger case. Perhaps on revising, the authors can make such a case.

Response to 7: This is a very interesting question that has been addressed in the conclusion section. The urgent need of medical solutions for these diseases, and the current use of nitro compounds with undeniable antitrypanosomal activity, namely benznidazole and nifurtimox, and more recently fexinidazole, justify the research in these family of compounds. Furthermore, DNDi-0690 is one of the molecules entering phase I clinical trial with nitroheterocyclic structure designed against visceral leishmaniasis. In our opinion, despite their relative intrinsic toxicity, the balance risk benefit, deserve the research on new nitro derivatives candidates.

Lastly, I emphasise that these are my personal reactions to reading the review. As I wrote at the top of this opinion, there is a lot of valuable information brought together. My concern is that its evaluation could be more helpful to readers.

Thank you for your valuable comments.

Reviewer 2 Report

This is an exaustive review on the chemotherapy against diseases aused by membersr of the Trypanosomatidade, especially based on the use of nitroheterocyclic compounds. 206 references are listed. However, there are some that should be mentioned, as listed below.

1. In relation to benznidazole there is a recent paper by the Navarro group describing interesting results on the silver and copper-benzinidazole derivatives against T. cruzi published recently in J. Inorg. Biochem.;

2. There is also a recent work on the use of SQ109 published by Baek et al, in biomedicines 2022;

3. Padilla et al also descibed very important results using orally active benzoxaborole for Chagas Disease (Nature Microbiol. 2022);

4. Zuma et al. also reported effect of fexinidazole on T. cruzi (Scientific Report 2022);

5. I also missed references to a series of papers published in the last years with several phospholipid analogues.

6. In the introduction to Chagas disease it is important to mention that in some countries most of the human acute infections are due to oral infection due to the ingestion of T. cruzi containing juices fruits rather than the cassical direct vector transmission. 

Author Response

Response to Reviewer 2 Comments

This is an exhaustive review on the chemotherapy against diseases caused by members of the Trypanosomatidae, especially based on the use of nitroheterocyclic compounds. 206 references are listed. However, there are some that should be mentioned, as listed below.

Point 1. In relation to benznidazole there is a recent paper by the Navarro group describing interesting results on the silver and copper-benzinidazole derivatives against T. cruzi published recently in J. Inorg. Biochem.;

Response to 1: Thank you for the comment. We have added a paragraph about it.

“Novel metal (Cu and Ag)-benznidazole complexes have been recently synthesized and tested against T. cruziepimastigotes and amastigotes, with Ag-Benznidazole complexes showing better activity than benznidazole alone, and showing higher selectivity towards T. cruzi parasites (de Souza et al., 2023).”

Point 2. There is also a recent work on the use of SQ109 published by Baek et al, in biomedicines 2022;

Response to 2: Thank you for the comment. We think this compound does not fit the content of this review, since it is not a current treatment against trypanosomatids, it is not entering clinical trials against the target diseases and it is not a nitro-heterocyclic compound.

Point 3. Padilla et al also described very important results using orally active benzoxaborole for Chagas Disease (Nature Microbiol. 2022);

Response to 3: Thank you for the comment. We have added a paragraph about it.

“Recently, the benzoxaborole prodrug AN15368 has been reported to be effective in the treatment of T. cruzi infections in non-human primates. In addition, this compound is orally active and exhibited no overt toxicity in a 60 d course, thus pointing it as a strong candidate for future clinical trials against Chagas disease (Padilla et al., 2022).”

Point 4. Zuma et al. also reported effect of fexinidazole on T. cruzi (Scientific Report 2022);

Response to 4: Thank you for the comment. A paragraph has been added:

in the low micromolar range… “Fexinidazole has been reported to promote massive disorganization of reservosomes, the detachment of the plasma membrane, unpacking of nuclear heterochromatin, mitochondrial swelling, Golgi disruption and alterations in the kinetoplast-mitochondrion complex in T. cruzi (Zuma and de Souza, 2022).

Point 5. I also missed references to a series of papers published in the last years with several phospholipid analogues.

Response to 5: Thank you for the comment. We have included miltefosine due to its current use for the treatment of leishmaniasis. However, we have not included other phospholipid analogues, since to the best of our knowledge they are not in clinical trials against diseases caused by trypanosomatids or have nitro-heterocyclic structure. We believe that inclusion of more classes of compounds would make the article very long.

Point 6. In the introduction to Chagas disease it is important to mention that in some countries most of the human acute infections are due to oral infection due to the ingestion of T. cruzi containing juices fruits rather than the cassical direct vector transmission. Parasitic infections continue to be a problem world-wide and this review brings together a great deal of information relevant to existing and new treatments. To that extent, I have no doubt that it should be published and will be found useful by a broad scientific community. It is also appropriate to publish in the suggested journal issue. I should like to raise a number of points with the authors for consideration.

Response to 6: Thank you for the comment. We have included this information in a paragraph in the introduction to Chagas disease:

“In some countries oral transmission of Chagas disease has been proposed, the primary vehicle being food and beverages contaminated with whole infected triatomines or their faeces including metacyclic trypomastigotes of T. cruzi (Noya et al., 2015).”
